# DeepDFA: Dataflow Analysis-Guided Efficient Graph Learning for Vulnerability Detection

## Abstract

Deep learning-based vulnerability detection models have recently been shown to be effective and, in some cases, outperform static analysis tools. However, the highest-performing approaches use token-based transformer models, which do not leverage domain knowledge. Classical program analysis techniques such as dataflow analysis can detect many types of bugs and are the most commonly used methods in practice. Motivated by the causal relationship between bugs and dataflow analysis, we present DeepDFA, a dataflow analysis-guided graph learning framework and embedding that use program semantic features for vulnerability detection. We show that DeepDFA is performant and efficient. DeepDFA ranked first in recall, first in generalizing over unseen projects, and second in F1 among all the state-of-the-art models we experimented with. It is also the smallest model in terms of the number of parameters, and was trained in 9 minutes, 69x faster than the highest-performing baseline. DeepDFA can be used with other models. By integrating LineVul and DeepDFA, we achieved the best vulnerability detection performance of 96.4 F1 score, 98.69 precision, and 94.22 recall.

## 1 Introduction

Software vulnerabilities cause great harm to people and corporations. Many Internet users have had their personal information breached because of security vulnerabilities, with common reports of breaches exposing millions of records (wik, 2021). The average data breach costs the target company $4.24 million, according to IBM's 2021 report (ibm, 2021). The number of vulnerabilities is growing every year, as reported by the Common Vulnerability Enumeration (CVE) from 2016-2021 (cve, 2021). Because of its importance, software companies invested heavily to develop vulnerability detection tools that can scan software before its release (Lu et al., 2021; Zheng et al., 2021).

Deep neural networks have reported great progress for vulnerability detection, with the recent LineVul paper (Fu & Tantithamthavorn, 2022) reporting 0.91 F1 score on a commonly used real-world vulnerability dataset (Fan et al., 2020), and many deep learning-based tools outperforming static analysis (Li et al., 2018; Ding et al., 2022; Cao et al., 2022). Current state-of-the-art models use graph neural networks (GNNs) with unsupervised word embeddings and large pretrained transformers, which can perform well on this task. However, these models are large and expensive to train, and we showed that they did not generalize well beyond unseen projects. Empirical studies have discovered that the models can focus on spurious features which are not relevant to the cause of the bug, such as variable names (Chakraborty et al., 2021).

Inspired by the work done by Xu et al. (2019) and by Cranmer et al. (2020), we designed a novel graph learning framework and embedding technique that is guided by program analysis algorithms of vulnerability detection, namely *Dataflow Analysis (DFA)*. DFA computes the data usage patterns and relations in the control flow graph (CFG) of a program and reports a vulnerability based on its root cause, i.e., whether the values and data relations collected from the program indicate the occurrence of the vulnerable conditions. We explored an analogy between DFA and the GNN message-passing algorithm, and designed an embedding technique that encodes dataflow information at each node of the CFG. Graph learning on such embedding thus simulates the dataflow computation in DFA.

We propose DeepDFA, *a Deep Learning Framework guided by DFA*, shown in Figure 1. Given the source code of a potentially vulnerable program, we convert it to a CFG and encode the nodes

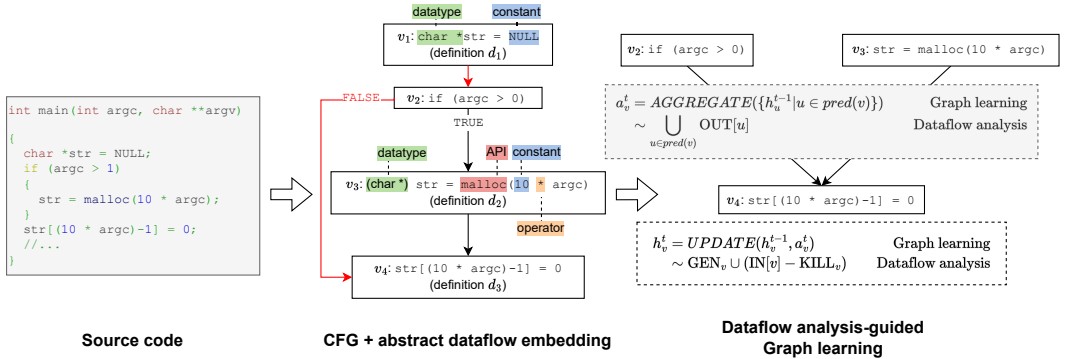

Figure 1: Overview of our DeepDFA approach

using an *abstract dataflow embedding* we designed. Our abstract dataflow embedding represents variable definitions using the properties which are most important for vulnerability detection, based on domain knowledge from program analysis, e.g., the data types, API calls, constants, and operators present in the definition. Then, we apply graph learning and its message passing mechanism on the CFG edges to propagate such information across the edges of the graph, similar to what is done in a dataflow analysis. We then use the learned graph representation to classify whether the function is vulnerable or not.

Our evaluation shows that DeepDFA is significantly faster than our baseline models, both in terms of training and inference time. It only took 9 minutes to train and 1.92 ms/example for inference on CPU. Yet, DeepDFA still achieved top one for recall, ranked second for F1 and generalized to unseen projects the best, compared with other baseline models. Importantly, we show that DeepDFA embedding can be used with other models to further improve their performance. We created the top state-of-the-art vulnerability detection model by combining LineVul and DeepDFA and achieved 96.40 F1 score, 98.69 precision and 94.22 recall.

Along the research like GIN Xu et al. (2019) and Cranmer et al. (2020), DeepDFA also demonstrated that guided by domain algorithms, deep learning can achieve better results with minimal resources. DeepDFA did not use any token and text level features which hardly reveal the cause of vulnerabilities; instead it used abstract dataflow embedding to simulate the dataflow propagation (a casual domain algorithm) via graph learning, and used the idea of bitvector in dataflow analysis to achieve the efficiency of embedding and learning.

In summary, we made the following contributions in this paper:

1. We designed an abstract dataflow embedding to efficiently learn the program semantics relevant to vulnerability detection;

2. We applied graph learning on the control flow graph (CFG) of the program and abstract dataflow embedding to simulate reaching definition dataflow analysis;

3. We implemented DeepDFA and experimentally demonstrated that DeepDFA outperforms baselines in vulnerability detection for effectiveness, efficiency, and generalization over unseen projects;

4. We provided a comprehensive understanding on the analogy of dataflow analysis and graph learning, which can understand why DeepDFA performs well and is efficient; and

5. We showed that DeepDFA can be used to improve other models, and we delivered the best vulnerability detection model in the state-of-the-art by combining LineVul and DeepDFA.

## 2 RELATED WORK

A vulnerability is a flaw in a software program that can be exploited to negatively impact the consumer or system (cve, 2022). In the literature, vulnerability detection is typically framed as a binary classification problem. Devign (Zhou et al., 2019), ReVeal (Chakraborty et al., 2021), IVDetect (Li

et al., 2021a), and LineVD (Hin et al., 2022) used GNN on program graph representations such as AST, CFG, and PDG, and annotated the nodes with unsupervised or pretrained word embeddings. ReGVD (Nguyen et al., 2022) used GNN on co-occurrence graphs. We also used a graph neural network, but we designed our graph learning framework using dataflow analysis as guidance. We therefore are able to understand why our approach is efficient and performant.

Transformer models such as CodeBERT (Feng et al., 2020) and LineVul (Fu & Tantithamthavorn, 2022) used a token-based program representation pretrained on a large body of NL-PL pairs, and then fine-tuned for vulnerability detection. Using CFGs, our graph learning only propagates the information along semantically important edges instead of trying to learning the relations of each pair of tokens. Thus, our approach is significantly more efficient. Since we have used a semantic-based embedding, we show that we can improve the performance of token based models. The most recent work, ContraFlow (Cheng et al., 2022), learns embeddings of def-use paths (an output of dataflow analysis), then predicts vulnerability detection using a transformer model. Our work directly emulates dataflow analysis and does not require an expensive pretraining phase.

There are also models that used sequence and CNN architectures. VulDeePecker (Li et al., 2018) used BiLSTM on slices considering data dependencies. SySeVR (Li et al., 2021b) used BiGRU on slices and adds data dependencies. Draper (Russell et al., 2019) used CNN and RF on a grid representation of source code.

While we were developing our analogy of DFA and graph learning, we found that Cummins et al. (2021) have proposed a similar idea, but used it for device mapping and algorithm classification. However, their PrograML model used an unsupervised embedding which does not explicitly capture the domain-specific information used by DFA, and they did not target a specific dataflow analysis. Our abstract dataflow embedding captures the domain-specific information used by DFA and we applied reaching definition analysis, which is directly linked to vulnerability detection.

Other relevant work that explores dataflow analysis and deep learning include: (1) VenkataKeerthy et al. (2020) used the output of dataflow analysis, reaching definitions and live variables, to learn flow-aware embeddings; and (2) Bielik et al. (2017) and Jeon et al. (2019) learned static analysis formulas from a dataset based on a fixed language. None of these works aims to develop a model for vulnerability detection.

## 3    DEEPDFA: EFFICIENT GRAPH LEARNING VIA DOMAIN GUIDANCE

Inspired by Xu et al. (2019) and Cranmer et al. (2020), we aim to design a graph learning framework that can simulate and thus benefit from the domain algorithms for vulnerability detection.

### 3.1    DATAFLOW ANALYSIS FOR VULNERABILITY DETECTION

Dataflow analysis (DFA) is a method for computing data usage patterns in a program. In addition to compiler optimization, dataflow analysis is an important method for vulnerability detection. One instance of dataflow analysis, called *reaching definition analysis*, reports at which program points, a particular variable definition can *reach*. A definition reaches a node when there is a path in the CFG that connects the definition and the node, and the variable is not redefined along the path. The reaching definition analysis can detect a null-pointer deference vulnerability based on its root cause, when it identifies that a definition of an NULL pointer reaches a dereference of the pointer. Similarly, it is a causal step to detect many other vulnerabilities such as buffer overflows, integer overflow, uninitialized variables, double-free and use-after-free (Cesare, 2013).

DFA uses two equations to propagate the dataflow information through the neighbouring nodes in the CFG, namely *meet-over-path* and *transfer function* (Aho & Aho, 2007). Meet-over-path aggregates the dataflow sets from its neighbours. Transfer function updates the dataflow set using the information available in node $v$. In the reaching definition analysis, the dataflow set is a set of definitions that reach a program point. A simple approach of performing a DFA is the *Kildall method* (Kildall, 1973) (the full algorithm is given in Appendix B). It iteratively propagates the dataflow information to the neighbours of $v$ in the CFG, a step at a time. The algorithm terminates when the dataflow information of all nodes stops changing, denoted a *fixpoint*. At termination, all nodes will incorporate the dataflow information from all other relevant nodes. This information is

then compared to the vulnerability conditions to determine whether a vulnerability has occurred in the program.

## 3.2 ANALOGY OF GRAPH LEARNING AND DATAFLOW ANALYSIS

Figure 2a shows a snippet of CFG where DFA is performed. In the CFG, each node is a statement, and each edge indicates the order of execution between two statements. The two dataflow equations that define a reaching definition dataflow analysis are shown in the figure:

- meet-over-path: $IN[v] = \bigcup_{u \in pred(v)} OUT[u]$
- transfer function: $OUT[v] = GEN_v \bigcup (IN[v] - KILL_v)$

where $IN[v]$ and $OUT[v]$ are the sets of dataflow located at the beginning and end of a statement. $GEN_v$ and $KILL_v$ represent the dataflow *generated* (new definitions) and *killed* (overwritten definition) in node $v$.

Figure 2b shows an analogous behavior of graph learning. Graph learning performs a fixed number of iterations of message-passing. At each iteration, each node aggregates information from its neighbors, and then updates its state to integrate the information. Here, the AGGREGATE and UPDATE functions serve a similar role to the two dataflow equations of meet-over-path and transfer function:

- aggregate: $a_v^t = AGGREGATE(\{h_u^{t-1} | u \in pred(v)\})$
- update: $h_v^t = UPDATE(h_v^{t-1}, a_v^t)$

where $a_v^t$ denotes the aggregated information from the neighbors, $h_v^t$ denotes the state of node $v$ after $t$ iterations of message-passing (analogous to $OUT[v]$).

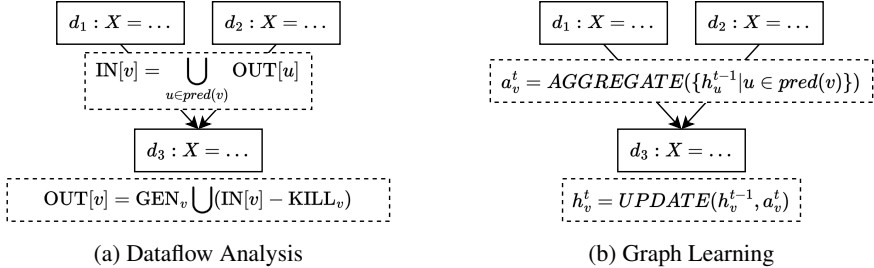

(a) Dataflow Analysis          (b) Graph Learning

Figure 2: Analogy of Dataflow Analysis and Graph Learning

## 3.3 GRAPH LEARNING TO SIMULATE DATAFLOW ANALYSIS

Based on the analogous behaviors of DFA and GNN, we developed DeepDFA. It conducts graph learning on the CFG of a program, guided by the traditional DFA algorithm. In addition, we designed a node embedding that can represent the dataflow set at each node.

**Designing Abstract Dataflow Embedding**: In dataflow analysis, we use a *bit vector* to represent the dataflow set at each node. A bit vector consists of $n$ bits of 0s and 1s. Its length is the size of the domain. A bit is set to 1 if its corresponding element is present in the set. In reaching definition analysis, the domain consists of all the definitions in the program, and the bits are set to "1" if the corresponding definitions reach the node. For example, in Figure 1, the program contains three definitions at nodes $v_1$, $v_3$, and $v_4$ so the reaching definition analysis uses a bit vector [0 0 0] to initialize each node before the analysis. The bit vector represents OUT[$v$] in the dataflow equation. It is updated at each step of propagation, and when the analysis terminates, the bit vectors for each node represent all possible definitions which reach that node.

The bit vector representation efficiently encodes program semantic features relevant to vulnerability detection. It can be quickly obtained at the node via lightweight program analysis. However, in graph learning, we cannot directly use it as the node embedding for two reasons. First, the domain of definitions is specific to a program. Different programs contain different variable definitions; their

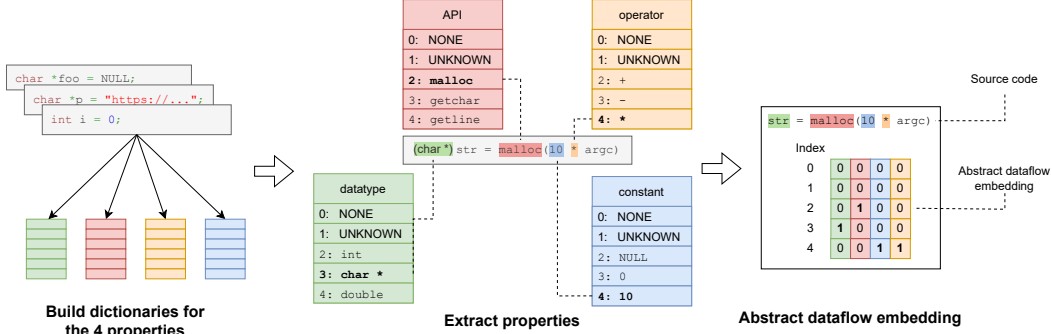

Figure 3: Abstract dataflow embedding generation

bit vectors have different lengths and the elements are not comparable. In graph learning, we want to extract dataflow patterns from all the programs and we need to have a "global" embedding that can be compared across programs. Second, bit vectors are changed in a discrete fashion during dataflow analysis; however, during graph learning, the node embedding needs to be changed in continuous increments.

To address the challenges, we map all the concrete definitions in the programs in the training dataset to *abstract definitions* using the common properties of the definitions. Thus, the definitions from different programs can be compared and generalized. Using domain knowledge, we believe that the following four properties are important to the cause of a vulnerability and can be shared across programs, so we use them to represent a definition:

- API call: the call to library or system functions used in the definition, e.g. `malloc`, `strlen`
- Data type: the data type of the variable being assigned, e.g. `int`, `char*`.
- Constant: the constant values assigned in the definition, e.g. `NULL`, `-1`, `"foo"`.
- Operator: the operators used in the definition, e.g. `+`, `-`, `*`.

We analyze a large corpus of programs, e.g., training set, and collect top $k$ frequently used API calls, data types, constants and operators to construct the dictionaries. For a particular definition, we identified its API call, data type, constant and operator, found their indices in the dictionaries, and converted each index to a vector using *one-hot encoding*. The generated embedding is a matrix of $k$ rows and 4 columns (each column is a one-hot vector representing one of the 4 properties), which we name *Abstract Dataflow Embedding*. Note that $k$ is a hyper-parameter of DeepDFA that we can tune. Using larger $k$, we can represent more definitions using our embedding precisely, and of course, with larger overhead. If the property's value was not in the top-$k$, we used an UNKNOWN token to represent its value; if the property is not present in the definition, we used a NONE token.

Figure 3 describes an abstract dataflow embedding for an example $d_2$: `str = malloc(10 * argc)`, which used an API call `malloc`, with the constant `10`, operator `*` and data type `char*`. Contrasted with the 3-bit bit vector that represents a concrete definition in data flow analysis, the abstract embedding is larger (5x4 elements), but it is still a fixed size, so it will scale to any size of dataset without impacting the model's efficiency. The embedding keys have consistent meanings among all programs in the dataset so the model can capture the dataflow patterns across programs.

**Graph learning guided by Dataflow Analysis:** Given the CFG with abstract dataflow embeddings, DeepDFA performs message-passing on the CFG edges, which parallels the dataflow propagation in the DFA algorithm.

In Table 1, we stepped through the reaching definition analysis for the CFG example in Figure 1 in order to further demonstrate the analogy between the two, and to help understand how our graph learning framework can benefit from the DFA guidance: (1) we only focus on the semantic information relevant to the vulnerability (2) the rules are simple and thus possible to learn (3) the iterative propagation can integrate information along the paths of a CFG to detect bugs based on its causes.

The row *Iteration 0* shows the initialization of each node in the reaching definition analysis. At iteration 1, the DFA updates OUT[$v_1$], OUT[$v_3$] and OUT[$v_4$] using the transfer function to indicate that the new definitions are introduced at the nodes. At iteration 2, OUT[$v_1$](including $d_1$) propagates to $v_2$ and OUT[$v_3$](including $d_2$) propagates to $v_4$, through the CFG edges. At iteration

Table 1: OUT[$v$] at each iteration of DFA

| Iteration | $v_1$ | $v_2$ | $v_3$ | $v_4$ |
|---|---|---|---|---|
| 0 | [0 0 0] | [0 0 0] | [0 0 0] | [0 0 0] |
| 1 | [1 0 0] | [0 0 0] | [0 1 0] | [0 0 1] |
| 2 | [1 0 0] | [1 0 0] | [0 1 0] | [0 1 1] |
| 3 | [1 0 0] | [1 0 0] | [0 1 0] | [1 1 1] |

3, the meet-over-path is used to combine OUT[$v_2$] and OUT[$v_3$]. Specifically, IN[$v_4$] $=$ $\bigcup\{$OUT[$v_2$], OUT[$v_3$]$\}$, computed as [1 0 0] $\lor$ [0 1 0] $=$ [1 1 0]; then the transfer function combines IN[$v_4$] with $GEN_{v_4}$, resulting in OUT[$v_4$] $=$ [1 1 1].

In graph learning, the analogous functions of AGGREGATE and UPDATE are learned using the neural networks. For example, in GGNN (Li et al., 2016), AGGREGATE is an MLP and UPDATE is a GRU. AGGREGATE combines the representations of multiple neighboring nodes into one vector, similar to meet-over-path. UPDATE uses GRU to preserve or forget portions of the previous state $h_v^{t-1}$ and integrate new information from $a_v^t$, similar to the operations performed in the transfer function. Within 2 iterations, $v_4$ will integrate the information from nodes $v_2$ and $v_3$ (i.e., the definitions of $d_1$ and $d_2$ respectively) with its initial dataflow information $d_3$.

After the DFA algorithm terminates, the final state of the nodes is used to detect vulnerabilities. The state of $v_4$ is [1 1 1], which indicates that both $d_1$ and $d_2$ may reach $v_4$ depending on the program values. Because the definition $d_1$ : str = NULL can reach the dereference at $v_4$, we can conclude that this program has a null-pointer dereference vulnerability. Similarly, in graph learning, after a fixed number of iterations, all the node representations are combined using a graph readout operation to produce a graph-level classification for vulnerability detection.

As Cummins et al. (2021) noted, one minor difference between the two algorithms is that DFA iterates until it reaches a fixpoint, while GNN performs a fixed number of iterations. Kam & Ullman (1977) showed that DFA converges within $d(G) + 3$ passes, where $d$ is the *loop connectedness* of $G$ (defined in Cummins et al. (2021)). We set the hyperparameter $t$ based on the best validation performance through our experimentation.

## 4 EVALUATION

We studied the 3 research questions: (1) Is DeepDFA **effective** for finding vulnerabilities? (2) How **efficient** is DeepDFA compared to other models? (3) Can DeepDFA **generalize** to unseen projects?

### 4.1 EXPERIMENT SETUP

**Model Implementation:** To explore whether DeepDFA can advance the state-of-the-art, we created two settings, **DeepDFA** and **DeepDFA+LineVul**. DeepDFA is implemented using the GGNN architecture (Li et al., 2016). DeepDFA+LineVul is created by concatenating LineVul's RoBERTa encoder output to DeepDFA's graph stage output after readout. To avoid data leakage, we trained the abstract dataflow embedding on the training set. We used a threshold of $k = 1000$, and it has covered (79.38%) of the definitions in the test dataset. Future work may train the abstract dataflow embedding on a large unsupervised dataset of code in order to learn a more general representation. See Appendix A for the further details of our hyperparameters and training.

**Dataset:** In the recent literature (Fu & Tantithamthavorn, 2022; Li et al., 2021a; Chakraborty et al., 2021), vulnerability detection models are typically evaluated with Devign (Zhou et al., 2019) or Big-Vul (Fan et al., 2020) datasets, both of which contains real-world open-source C/C++ projects. In our evaluation, we used the Big-Vul dataset because (1) it is bigger compared to Devign and consists of 188,636 functions, and 10,900 (6%) vulnerable labels, and (2) it reflects the imbalanced distribution of real-world code (Devign is a balanced dataset), with the minority of code containing vulnerabilities (Chakraborty et al., 2021).

We used the train/validation/test splits of 80/10/10% generated by the LineVul paper (Fu & Tantithamthavorn, 2022). To address class imbalance, we undersampled the majority class (non-

vulnerable) of the training set without changing the validation and test sets. We used Joern (Yamaguchi et al., 2014)[1] to parse the code into its CFG representation. Because Joern cannot parse some programs (7%) in the dataset, we used the rest of the 93% of the data in our experiments.

We also created a datasest to evaluate the generalization of the models. This dataset consists of the *mixed-project* and *cross-project* two settings. Our approach is to hold out 10k examples from randomly selected projects in Big-Vul as a validation and test set of the *cross-project* setting. We then randomly partitioned the remaining data into train/validation/test sets with 10k examples in the validation and test data, and the remaining examples in training. The second 10k examples can come from the same projects as the examples used in training, which we call the *mixed-project* setting. To mitigate the potential bias caused by the selection of projects, we repeated this process 5 times with different selections of the cross-project data and performed 5-fold cross validation.

**Baselines:** We compared against 7 non-transformer models: VulDeePecker, SySeVR, Draper, Devign, ReVeal, ReGVD, IVDetect, and 2 transformer models: CodeBERT and LineVul[2]. These models are developed recently with different architectures and represent the state-of-the-art of vulnerability detection models. See Section 2 for an overview of the models and Appendix A for the details of our reproductions.

**Metrics:** We evaluated model performance and efficiency using metrics from the literature. Specifically, **Precision** reports the portion of positive predictions which were correct: $P = \frac{TP}{TP+FP}$. **Recall** calculates the portion of positive examples which were recalled: $R = \frac{TP}{TP+FN}$. **F1** is the harmonic mean between Precision and Recall: $F_1 = 2 * \frac{P*R}{P+R}$. We used $F_1$ to decide the highest performing model because it balances precision and recall, which are both important in an imbalanced dataset.

For efficiency, we reported the **Training time**: the wall-clock time to execute one training run with one validation run per epoch, and the **Inference time**: the average wall-clock time to predict for one example. We also used **MACs**: the average number of Multiply-Accumulate operations[3] (Tan & Le, 2020; Howard et al., 2017) to predict for one example; this measures the performance independently of the computing platform. Finally, we listed the model size in terms of **Parameter count**: the number of trainable parameters in the neural network model.

We trained the models 3 times with different random seeds and reported the mean score and standard deviation for each metric. We ran the runtime benchmarks on an AMD Ryzen 5 1600 3.2 GHz processor with 48GB of RAM and an Nvidia 3090 GPU with 24GB of GPU memory.

## 4.2 RESULTS: EFFECTIVENESS

**Comparison with non-transformer models:** As shown in Table 2a, DeepDFA performed significantly better than the baselines on F1 score; the average difference was 47.51. DeepDFA also achieved a significantly higher Recall score compared to other models, and its Precision score ranked 4th out of the 7 baselines. Compared to the other 6 models, DeepDFA reported significantly lower variances for all the three metrics.

DeepDFA's abstract dataflow embedding encodes the semantic features (dataflow sets) of programs relevant to the vulnerability detection. On the other hand, the other models used unsupervised word embeddings such as Word2Vec, Doc2Vec, and GloVe as node embedding, which do not represent a causal relationship with vulnerabilities. Among the 7 baseline tools, Devign and ReVeal used CPG which combines AST, CFG, and PDG, while DeepDFA used only the CFG. AST and PDG information are useful to detect certain classes of bugs (Yamaguchi et al., 2014), but compared to DeepDFA which directly simulate the dataflow analysis on the CFG, these approaches are less accurate to model data relations needed for causal vulnerability detection.

**Comparison with transformer models:** CodeBERT and LineVul are the best among all the transformer models that detect vulnerabilities in the BigVul dataset (see Appendix C for the detailed numbers). Table 2b shows that DeepDFA reported the highest recall and the smallest variances compared to CodeBERT and LineVul. It ranked second for F1 and performed significantly better than CodeBERT in F1. We achieved the state-of-the-art performance of F1 score of 96.40 (3.17

---

[1] https://joern.io

[2] We could not reproduce LineVD and ContraFlow on Big-Vul for function-level vulnerability detection.

[3] We used DeepSpeed profiler to measure MACs. https://www.deepspeed.ai

Table 2: DeepDFA outperformed the baselines and can be used to further improve the existing model performance. Note that VulDeePecker, SySeVR, Draper, and IVDetect performance were directly taken from the IVDetect paper (Li et al., 2021a), so we do not report the variance.

(a) Comparison with non-transformer models.

| Model | F1 | P | R |
|---|---|---|---|
| VulDeePecker | 12.00 | 49.00 | 19.00 |
| SySeVR | 15.00 | **74.00** | 27.00 |
| Draper | 16.00 | 48.00 | 24.00 |
| IVDetect | 23.00 | 72.00 | 35.00 |
| Devign | 26.85 | 29.00 | 25.03 |
| | (0.97) | (0.38) | (1.67) |
| ReVeal | 32.94 | 34.27 | 31.73 |
| | (0.75) | (1.58) | (0.65) |
| ReGVD | 19.15 | 63.67 | 11.33 |
| | (2.65) | (4.43) | (1.94) |
| DeepDFA | **68.26** | 53.98 | **92.81** |
| | (0.16) | (0.06) | (0.40) |

(b) Comparison with transformer models.

| Model | F1 | P | R |
|---|---|---|---|
| CodeBERT | 21.04 | 68.48 | 12.91 |
| | (6.72) | (11.76) | (5.51) |
| DeepDFA | 68.26 | 53.98 | 92.81 |
| | (0.16) | (0.06) | (0.40) |
| LineVul | 93.23 | 97.32 | 89.48 |
| | (0.31) | (0.66) | (0.42) |
| DeepDFA+LineVul | **96.40** | **98.69** | **94.22** |
| | (0.13) | (0.28) | (0.46) |

improvement), Precision score of 98.69 (1.37 improvement), and Recall score of 94.22 (4.75 improvement), by adding DeepDFA's embedding to LineVul. Since DeepDFA is lightweight, it can be added to improve LineVul at little cost. The improvement suggests that DeepDFA's embedding can provide additional useful information even for a very large pretrained models trained from the tokens of programs.

DeepDFA does not use any text-level/token information such as variable and function names, and yet it has achieved very good performance. We believe that leveraging the domain algorithm of reaching definition analysis to guide graph learning indeed plays an important role, and the embedding indeed encodes semantic features (e.g., data relations) that are important for vulnerability detection. Interestingly, such features can also improve the top-performing model along all metrics. We further believe that the examples which DeepDFA predicted incorrectly could be due to the fact that reaching definition analysis cannot handle all types of vulnerabilities. Thus, by adding other dataflow analyses such as live variable analysis, DeepDFA can further improve its performance. We will leave this step to our future work.

## 4.3 RESULTS: EFFICIENCY

Table 3: Runtime of the different models.

| Model | Train time (ms) | GPU Inference (ms/example) | CPU Inference (ms/example) | MACs/example |
|---|---|---|---|---|
| LineVul | 10h19m | 11.06 | 1068.2 | 48.32 B |
| DeepDFA | 9m | 4.64 | 5.8 | 40.27 M |
| DeepDFA+LineVul | 10h40m | 15.36 | 1571.5 | 48.32 B |

In Table 3, we present the runtime comparison of DeepDFA and LineVul. Here, we did not list other models because their performances are much worse (shown in Section 4.2), and they took hours to train, compared to DeepDFA which finished training in 9 minutes (excluding data preprocessing time). In Table 4, we also listed the sizes of the models in terms of the number of parameters.

Compared to LineVul, DeepDFA took 69x less time to train, had 2.4x faster inference on GPU and 185x faster inference on CPU, and used 333x fewer parameters. DeepDFA had the least number of parameters of all models, equal to 67% of the smallest model (ReVeal) and 0.3% of the largest baseline model (LineVul). These results consistently indicate that DeepDFA excels in its efficiency compared to other models. This is possible because DeepDFA is based on the dataflow analysis's compact representation—-bitvector, which represents semantic information in bits and thus is more efficient compared to tokenized strings. DeepDFA propagated information along only the domain-specific CFG edges, rather than associating every pair of tokens in an exhaustive fashion.

DeepDFA's short inference time due to a low number of MAC operations enables its use in non-GPU environments (which are common for software development) where larger models such as LineVul may not be easy to be deploy. DeepDFA's short training time enables techniques like per-project fine-tuning and hyperparameter tuning, which would be much more costly with LineVul's training time of over 10 hours. Because of DeepDFA's small parameter count, it is ideal for resource-limited computing platforms such as mobile devices, where large models cannot be used (Howard et al., 2017).

Table 4: Size of the models.

| Model | # parameters |
|---|---|
| LineVul | 125,238,531 |
| DeepDFA | 375,938 |
| DDFA+LV | 125,679,236 |
| IVDetect | 924,165 |
| Devign | 1,148,553 |
| ReVeal | 560,291 |
| ReGVD | 124,794,500 |
| CodeBERT | 124,646,401 |

## 4.4 RESULTS: GENERALIZATION

We compared the *cross-project* (shown as Cross F1) and the *mixed-project* (shown as Mixed F1) test F1 scores to evaluate the models' capabilities of generalizing over unseen projects. Table 5 presents the highest-performing baseline model LineVul compared to DeepDFA (the results of other models are available in Appendix D). Shown under Column $\Delta$ F1, DeepDFA reported the most consistent performance between cross-project and mixed-project settings. When applying to unseen projects, DeepDFA

Table 5: How do the models handle unseen projects? Note the performance drop ($\Delta$ F1) from the cross-project to mixed-project setting.

| Model | Mixed F1 | Cross F1 | $\Delta$ F1 |
|---|---|---|---|
| LineVul | 84.03 | 71.37 | -12.66 |
| DeepDFA | 70.49 | 68.58 | **-1.91** |
| DDFA+LV | **87.89** | **71.88** | -16.02 |

only dropped 1.91 (2.7%) F1 score, compared to 12.66 (15.1%) drop for LineVul and 16.02 (18.2%) drop for DeepDFA+LineVul. DeepDFA+LineVul reported the best performance for both the mixed-project and cross-project datasets, improving on LineVul's mean F1 score by 3.74 and 0.54 points respectively. Although DeepDFA did not rank the best in terms of the absolute values, its generalization was significantly better than the other models because its performance dropped the least from the mixed-project set to the cross-project set.

We believe that DeepDFA generalizes better because it does not rely on superfluous features that may exist at token and text level such as variable names and function names as reported by the previous research (Chakraborty et al., 2021). For example, LineVul takes as input tokens, which often differ between different projects and thus are hard to generalize their patterns. On the other hand, our abstract dataflow embedding encodes the usage patterns of commonly used API calls, operators, constants, and data types. Such patterns can be directly related to the cause of the vulnerabilities, and thus might help DeepDFA generalize better over unseen projects.

## 5 CONCLUSIONS AND FUTURE WORK

We propose DeepDFA, an efficient graph learning framework and embedding technique guided by dataflow analysis for vulnerability detection. Our abstract dataflow embedding incorporates domain knowledge and extract semantic features of data usage patterns of commonly used API calls, operations, constants, and data types that potentially capture the causes of the vulnerabilities. The graph learning emulates the Kildall method of dataflow analysis using the analogous message-passing algorithm. Our experimental results confirm that DeepDFA is performant and efficient, and generalizes better than other models. DeepDFA can be trained in 9 minutes and yield a smallest model compared to all other baseline models, yet still achieving the top ranking for recall and no. 2 ranking for F1. We show that DeepDFA can be used to improve other models. By combining LineVul and DeepDFA, we delivered the best vulnerability detection model in the state-of-the-art at 96.40 F1 score, 98.69 precision and 94.22 recall. In the future, we plan to incorporate the guidance of other dataflow analyses such as live variable analysis for graph learning. These dataflow analyses are also used for vulnerability detection(Cesare, 2013). We also plan to explore the application of DeepDFA to precisely pinpoint the vulnerability location at specific lines in the code.

## 6 REPRODUCIBILITY STATEMENT

All code and data used in this paper are available at `https://figshare.com/s/e7953b4d345b00990d17`, along with trained model weights and scripts to run the experiments as we ran them.

- Section 4.1 documents our benchmarking environment.
- Appendix A documents the hyperparameters we used for the GGNN model and abstract dataflow embedding.
- We used Joern version `1.1.1072` to parse the CFG, available at `https://joern.io`.

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

## A    TRAINING DETAILS

| Hyperparameter | Value |
|---|---|
| $\lambda$ (learning rate) | $1e^{-3}$ |
| $L_2$ weight | $1e^{-2}$ |
| $k$ (threshold) | 1000 |
| $t$ (number of GNN steps) | 5 |
| Hidden size | 32 |
| # output layers | 3 |
| Batch size | 256 |

Table 6: Hyperparameters used for training in RQ1.

We trained DeepDFA for 25 epochs and evaluated the checkpoint with the lowest validation loss. We listed our hyperparameters in Table 6.

We trained DeepDFA+LineVul with the same method as LineVul; we trained for 10 epochs and evaluated the checkpoint with the highest validation F1 score.

For the baselines,

- We could not reproduce VulDeePecker, SySeVR, Draper, or IVDetect, so we repeated the performances reported in Li et al. (2021a). Their measurements may vary slightly from our reproduction.

- We confirmed with Chakraborty et al. (2021) that our results fixed a data leakage bug in the original implementation, so the results we report may differ from the original paper.

- Zhou et al. (2019) did not release their model code, so we reproduced Devign from the third-party implementation released by Chakraborty et al. (2021) (`https://github.com/saikat107/Devign`).

- Hin et al. (2022) did not report function-level metrics for LineVD, and we could not reproduce their statement-level performance from their model code, so we did not compare with their approach.

- Cheng et al. (2022) did not release their model code and evaluated a different dataset than ours, so we did not compare with their approach.

# B    THE DATAFLOW ANALYSIS ALGORITHM

Algorithm 1 lists the generic Kildall method for dataflow analysis. The meet-over-path $\sqcap$ and transfer function $f$ must be specified in order to compute a specific dataflow analysis.

---

**Algorithm 1** The round-robin iterative algorithm for dataflow analysis (i.e. Kildall method)

---

> **Given:** graph $G = (V, E)$, meet-over-path $\sqcap$, transfer function $f$, initial states $P^0$
> **for** $v$ in $V$ **do**
>     $\text{OUT}[v] \leftarrow P^0[v]$                                         ▷ set initial state for each node
> **end for**
> $changed \leftarrow True$                               ▷ denotes whether $\text{OUT}[v]$ changed for any $v$
> **while** changed **do**
>     $changed \leftarrow False$
>     **for** $v$ in $V$ **do**
>         $\text{IN}[v] \leftarrow \sqcap_{u \in pred(v)} \text{OUT}[u]$                                    ▷ Do AGGREGATE/$\sqcap$
>         $\text{OUT}'[v] \leftarrow f(\text{IN}[v])$                                          ▷ Do UPDATE/$f$
>         **if** $\text{OUT}'[v] \neq \text{OUT}[v]$ **then**
>             $\text{OUT}[v] \leftarrow \text{OUT}'[v]$
>             $changed \leftarrow True$           ▷ If any $\text{OUT}[v]$ changed, iterate again; otherwise terminate
>         **end if**
>     **end for**
> **end while**

---

## C  EFFECTIVENESS RESULTS ON THE FULL BIG-VUL DATASET

Of the transformer models, we evaluated CodeBERT and LineVul in our experiment for Section 4.2.

Table 7: Initial trial run of performance on 100% of the Big-Vul dataset.

| Model type | Model | F1 | Precision | Recall |
|---|---|---|---|---|
| GNN | Devign | 29.33 (6.58) | 32.83 (5.55) | 26.59 (7.25) |
| GNN | ReVeal | 33.6 (0.69) | 33.08 (3.49) | 34.67 (3.8) |
| GNN | ReGVD | 24.49 (3.16) | 63.76 (3.54) | 15.26 (2.71) |
| Transformer | CodeBERT | 22.68 (8.12) | 67.79 (4.9) | 19.11 (3.39) |
| Transformer | LineVul | 91.58 (0.49) | 95.99 (0.85) | 87.55 (0.49) |
| Transformer | VulBERTaMLP (Hanif & Maffeis, 2022) | 1.75 (3.03) | 19.33 (33.49) | 0.92 (1.59) |
| Transformer | VulBERTaCNN (Hanif & Maffeis, 2022) | 10.59 (0) | 5.59 (0) | 100 (0) |
| Transformer | PLBART (Ahmad et al., 2021) | 25.35 (3.74) | 61.84 (6.54) | 16.18 (3.52) |

## D  CROSS-PROJECT EVALUATION RESULTS ON THE FULL BIG-VUL DATASET

We only evaluated LineVul in our experiment for Section 4.4 because its absolute performance was significantly better than the other models on our initial trial with 100% of the Big-Vul dataset (See Table 8).

Table 8: Initial trial run of cross-project evaluation with 100% of the dataset.

| Model | Mixed-project F1 | Cross-project F1 | Δ F1 |
|---|---|---|---|
| LineVul | 84.07 | 71.81 | -12.26 |
| Devign | 24.32 | 14.26 | -10.06 |
| ReVeal | 28.24 | 6.83 | -21.41 |
| ReGVD | 33.61 | 21.70 | -11.91 |
| CodeBERT | 24.14 | 4.98 | -19.16 |

