# OpenReview forum: "DeepDFA: Dataflow Analysis-Guided Efficient Graph Learning for Vulnerability Detection"
_ICLR.cc/2023/Conference — Submitted to ICLR 2023_

### Official Review · Reviewer_ePbc · 2022-10-23

**Confidence:** 4
**Correctness:** 3
**Technical Novelty And Significance:** 2
**Empirical Novelty And Significance:** 2
**Recommendation:** 3

**Clarity, Quality, Novelty And Reproducibility:**

- Clarity: This paper is easy to follow.

- Novelty: It seems that the main contribution is the proposed abstract dataflow embedding, which seems to be limited.

- Reproducibility: The data and code is available for reproduction.

- Quality: The idea seems interesting, but technical novelty for model is not enough.

**Strength And Weaknesses:**

Strength:

- The related work is comprehensive, though there are some missing related works, it still cover current the most deep learning techniques for vulnerability detection.
- The idea to compare data flow analysis to message passing in GNNs seems interesting.
- This paper is easy to follow.

Weaknesses:

- This paper uses data flow for vulnerability dection. I agree that there are many vulnerability types caused by data dependencies, however how to deal with other vulnerabilites that are caused by other dependencies such as control dependencies?

- Why choosing four properties such as API call, Data type, Constant and Operator? Are these properties enough for vulnerability detection? There appears to be insufficient evidence to justify this choice.

- What is the novelty of model except the designed the abstract dataflow embedding? It seems that this paper just replace the embedding layer in GNNs to the abstract dataflow embedding for vulnerability detection. Furthermore, since the dataflow embedding size can be controlled by the hyper-parameter k, hence the model size and parameters will denifitely smaller than others.

- Why choose GGNN for the model architecture? Can we use other GNN variants for the replacement.

- I suggest authors to combine Table 2 (a) and (b) for better presentation.


**Summary Of The Paper:**

This paper proposes a dataflow analysis-guided graph learning approach namely DeepDFA for vulnerability detection. It encodes the node from control flow graph using an abstract dataflow embedding. Based on the vectorized node embedding, it further applies GGNN on the CFG for message passing. Then a graph representation is used to classify whether a function is vulnerable or not. By the experimental results, it confirms that DeepDFA is significantly faster than the baseline models. Furthermore, it also confirms that the designed embedding approach is also general to other vulnerability detection models.

**Summary Of The Review:**

Although the idea to compare data flow analysis with GNNs is interesting, the technical novelty is not enough. Furthermore, the reason behind the embedding layer is also unclear.

---

### Official Review · Reviewer_EyDV · 2022-10-24

**Confidence:** 4
**Correctness:** 3
**Technical Novelty And Significance:** 2
**Empirical Novelty And Significance:** 2
**Recommendation:** 3

**Clarity, Quality, Novelty And Reproducibility:**

Clarity: the paper describes the core idea clearly.

Quality: the quality of the writing can be improved, e.g., the description of the dataflow analysis can be made clearer. For example, there are also some discrepancies between the paper's description and the public algorithm, e.g., each node in CFG is usually a basic block, not a single statement. The description should be improved to walk through the algorithm in more detail.

Novelty: the analogy of graph learning and dataflow analysis is novel, but the actual design based on this insight is not convincing (see summary of the review).

Reproducibility: the authors have provided experiment details to help reproduce the paper.

**Strength And Weaknesses:**

+ Design: analogies of the dataflow analysis algorithm and the message passing algorithm in graph neural networks are intuitive and novel.
+ Results: the dataflow embedding improves the efficiency, precision, and generalizability of graph neural networks in vulnerability detection.

- Unclear design choices of abstract dataflow embedding, in which the variable -- the key part in dataflow analysis - is not included.
- Evaluation should show the vulnerability types and how dataflow analysis identifies them.

**Summary Of The Paper:**

This paper proposes DeepDFA, an efficient graph neural network architecture based on the dataflow information of the program for vulnerable function detection. It develops the abstract dataflow embedding to represent the program. The key insight is that such representation enables the graph learning algorithm to behave similarly to the dataflow analysis process, a classical static analysis approach to detect vulnerabilities. Experimental results demonstrate that DeepDFA is fast in training and inference and achieves the best results when combined with an existing baseline.

**Summary Of The Review:**

I like the idea of this paper that incorporates the dataflow analysis as the inductive bias of neural networks for program analysis. However, I think the paper needs to address the following concerns.

It is unclear why the properties used to represent each node in the graph mismatch the information used in actual dataflow analysis. As described in Section 3.3, the authors used 4 properties of a node without including the variables in the program, but they are critical to dataflow analysis. Without incorporating variables, it is hard to argue that the graph learning algorithm is still learning dataflow analysis.

Similar to variable names, which the authors discard as they are program-specific, the 4 properties used to represent statement nodes are also domain-specific. API calls, type strings, constants, and operators might change across different programming languages, and their distribution might vary significantly across projects. How do these properties generalize to unseen ones? Are there experiments to ablate their effectiveness?

What is the overhead of extracting the extra information of the program, i.e., abstract dataflow embedding, over using raw code tokens? It is helpful to show the tradeoff of increased data processing effort for improved training/inference efficiency.

Dataflow analysis algorithm assumes termination when reaching a fixed point. The authors also mentioned that the number of iterations depends on the graph size (d(G)+3). Why did the authors decide to fix such hyperparameters based on a validation set?

What are the vulnerability types in the evaluated dataset -- is dataflow analysis indeed enough to analyze all the vulnerability types? If yes, as a baseline, the authors should report the performance of the classic dataflow analysis and argue why we need an alternative based on machine learning. If not, it is important to show in which case dataflow analysis fails. How does the model perform in these cases? What makes the model outperform the standard dataflow analysis (if this is true), if it simply learns the dataflow analysis?

Minor:

The scale of the metrics described and used in the paper should be consistent. For example, the authors use 0.91 F1 scores in the introduction but use 96.4 F1 scores in the other places.

The authors of the citation "Compilers: principles, techniques, & tools" is incorrect.

Section 2, I do not think that "propagates the information along semantically important edges" are realized in this paper. This paper proposes to propagate information on control flow graphs, but only control flows do not fully capture all the dependencies, e.g., data dependencies. I am not fully convinced that such a setup is "significantly more efficient" than transformers.

---

### Official Review · Reviewer_x4r7 · 2022-10-24

**Confidence:** 4
**Correctness:** 3
**Technical Novelty And Significance:** 2
**Empirical Novelty And Significance:** Not applicable
**Recommendation:** 3

**Clarity, Quality, Novelty And Reproducibility:**

Clarity: The paper is easy to follow.
Novelty: The authors explored to use graph learning for vulnerability detection, but the novelty is minor.
Reproducibility: The data and code is available for reproduction.
Quality: The idea seems interesting, but novelty is not enough.

**Strength And Weaknesses:**

Strength:
The paper is easy to follow in general.
Vulnerability detection using graph learning is an important problem.

Weaknesses:
The authors explored to use graph learning for vulnerability detection, but the novelty is minor:
(1)The analogy of dataflow analysis and graph learning is somewhat trivial especially when there are already several existing works, for example, “ProGraML: A Graph-based Program Representation for Data Flow Analysis and Compiler Optimizations. ICML 2021” as this paper cited.

(2)Moreover, the dataflow embedding in this paper, which only select API calls, data type, Constant, and Operator, seems not strong enough to represent dataflow. There is also other more important information, for example, tainted value or expressions, path constraint, and so on. Take the example in Figure 1, as the authors explained at the end of section 3.3, they show a null-pointer dereference vulnerability because the definition d1:str=NULL can reach the dereference at v4. How about when there is an if statement that checks whether str == NULL, if it does, program will terminate. Will DeepDFA also report a null-pointer dereference vulnerability since path constraints are omitted in the dataflow embedding?

(3)The evaluation also should be improved, both in setting and description.

(3.1) I wonder what level and what kinds of vulnerability DeepDFA is able to detect?
In evaluation, the authors did not compare with ProGraML, which is the most related to this work. Although ProGraML was evaluated on device mapping and algorithm classification, ProGraML is also able to do data flow tasks (referring to ProGraML’s evaluation), such as Reachability analysis, Dominance analysis, liveness analysis, and so on. Since the two works are so related, it would be better to compare them in Reachability analysis, Dominance analysis, liveness analysis, and so on. If possible, also compare the ability of vulnerability detection by self-implementation based on ProGraML.


(3.2) As I know, CodeBERT is a model for natural language code search and code documentation generation, how the authors apply CodeBERT for vulnerability detection, and what kinds of vulnerability can be detected? It would be better to give more details.


(3.3) In the evaluation, the authors only show F1 score, precision, recall. What kinds of vulnerabilities are detected in the experiments? It would be better to show more details of the vulnerabilities.


(3.4) I noticed there are some closely related works not included and discussed:
“Learning to Represent Programs with Graphs, ICLR2018”
"Learning semantic program embeddings with graph interval neural network. OOPSLA 2020”



**Summary Of The Paper:**

In this work, the authors present DeepDFA, a dataflow analysis-guided graph learning framework and embedding that use program semantic features for vulnerability detection. In experiment, they show DeepDFA ranked first in recall, first in generalizing over unseen projects, and second in F1 among all the state-of-the-art models they experimented with.

**Summary Of The Review:**

The authors explored to use graph learning for vulnerability detection, but the novelty is minor:
(1)The analogy of dataflow analysis and graph learning is somewhat trivial especially when there are existing works, for example“ProGraML: A Graph-based Program Representation for Data Flow Analysis and Compiler Optimizations. ICML 2021” as this paper cited.
(2)Moreover, the dataflow embedding in this paper, which only select API calls, data type, Constant, and Operator, seems not strong enough to represent dataflow.
(3)The author shows experiment details, but can be improved

---

### Official Review · Reviewer_FYuz · 2022-10-24

**Confidence:** 4
**Correctness:** 3
**Technical Novelty And Significance:** 2
**Empirical Novelty And Significance:** 2
**Recommendation:** 3

**Clarity, Quality, Novelty And Reproducibility:**

Clarity: The overall end-to-end detection method could be described more clearly. More specifically, the granularity of detection and how to localize vulnerable code (if possible) in the proposed approach  is not clear from the description. Some clarifying questions on evaluations:
- Evaluation results in Table 2(b) shows that although DeepDFA performs poorly, concatenating its output to LineVul’s encoder output improves detection performance of LineVul. Is there any pattern  (e.g., type of vulnerability) which are missed by original LineVul, but detected by LineVul+DeepDFA?
- How does the performance of LineVul+DeepDFA compare with concatenation of LineVul with other similar GNN based approaches (e.g., Devign)?
- How to determine k? Effective k values could be different for different properties (e.g, api call vs data type).


Novelty: DeepDFA proposes encoding of dataflow analysis in CFG for vulnerability detection. However, the novelty of this work is not well-demonstrated. Although not specifically applied to vulnerability detections, PrograML has already used a similar approach for program representations.

Reproducibility: Code, data, trained models, and relevant models used in the paper are shared. Thank you!

**Strength And Weaknesses:**

Strengh

Dataflow analysis has been used as a useful tool for vulnerability analysis. Augmenting dataflow analysis to improve performance of ML based vulnerability detection approaches seems reasonable.

Weakness

- The use of GNN to model CFG of programs for vulnerability detection has already been explored by previous works (as mentioned in the related work) . So, the novelty of this work is restricted to ‘encode dataflow analysis in CFG’ for graph learning. However, it is not clear how (or if) the proposed embedding is more effective than the already proposed embedding approach in PrograML. There is no comparative discussion/evaluation that demonstrates the superiority of the proposed approach.
- It is not clear if the proposed approach detects vulnerabilities at the program/function/line level. Baseline methods seem to include detectors that focus on both function and line level.


**Summary Of The Paper:**

This paper proposes a GNN based approach for vulnerability detection. The control flow graph (CFG) of a program is augmented with dataflow information and graph learning is applied to predict vulnerability of the program. This work proposes an “abstract dataflow embedding” to represent important dataflow properties that are relevant to vulnerability detection and uses message-passing of the GNN model to mimic the dataflow analysis algorithm.  Evaluation of the proposed model on publicly available dataset shows improved performance compared to several state-of-the-art approaches.

**Summary Of The Review:**

This work inspires the use of dataflow information in GNN based vulnerability detection. However, the proposed approach has reduced novelty. Also, the evaluation and description of the overall system lacks clarity.

---

### Decision · Program_Chairs · 2023-01-20

**Decision:**

Reject

**Justification For Why Not Higher Score:**

Would like to see at least one reviewer champion acceptance or a very strong author response.

**Justification For Why Not Lower Score:**

N/A

**Metareview: Summary, Strengths And Weaknesses:**

Reviewers are in agreement that the paper isn't ready for publication in its current form, and there was no author response. There are a number of issues raised in the reviews, but the most pertinent are limited novelty and that reviewers were unconvinced by several of the design decisions that went into the model.